# Data from the German TwinLife Study: Genetic and Social Origins of Educational Predictors, Processes, and Outcomes

Journal of
open psychology data

DATA PAPER

]u[ ubiquity press

**THERESA ROHM** (iD)

**ANASTASIA ANDREAS** (iD)

**MARCO DEPPE** (iD)

**HARALD EICHHORN** (iD)

**JANA INSTINSKE** (iD)

**CHRISTOPH H. KLATZKA** (iD)

**ANITA KOTTWITZ** (iD)

**KRISTINA KRELL**

**BASTIAN MÖNKEDIEK** (iD)

**LENA PAULUS** (iD)

**SOPHIA PIESCH** (iD)

**MIRKO RUKS** (iD)

**ALEXANDRA STARR** (iD)

**LENA WEIGEL** (iD)

**MARTIN DIEWALD** (iD)

**CHRISTIAN KANDLER** (iD)

**RAINER RIEMANN** (iD)

**FRANK M. SPINATH** (iD)

*Author affiliations can be found in the back matter of this article

**CORRESPONDING AUTHOR:**

**Theresa Rohm**

University of Bremen, DE

rohm@uni-bremen.de

**KEYWORDS:**
extended twin family study; cross-sequential design; genetic and environmental factors; social inequality; educational differences

**TO CITE THIS ARTICLE:**
Rohm, T., Andreas, A., Deppe, M., Eichhorn, H., Instinske, J., Klatzka, C. H., Kottwitz, A., Krell, K., Mönkediek, B., Paulus, L., Piesch, S., Ruks, M., Starr, A., Weigel, L., Diewald, M., Kandler, C., Riemann, R., & Spinath, F. M. (2023). Data from the German TwinLife Study: Genetic and Social Origins of Educational Predictors, Processes, and Outcomes. *Journal of Open Psychology Data*, 11: 4, pp. 1–15. DOI: https://doi.org/10.5334/jopd.78

## ABSTRACT

The major aim of the German TwinLife study is the investigation of gene-environment interplay driving educational and other inequalities across developmental trajectories from childhood to early adulthood. TwinLife encompasses an 8-year longitudinal, cross-sequential extended twin family design with data from same-sex twins of four age cohorts (5, 11, 17, and 23 years) and their parents, as well as their non-twin siblings, partners, and children, if available, altogether containing $N = 4,096$ families. As such, TwinLife includes unique and openly accessible data that allows, but is not limited to, genetically informative and environmentally sensitive research on sources of inequalities regarding educational attainment, school achievement, and skill development.

# 1 BACKGROUND

The purpose of the German Twin Family Panel (TwinLife) encompasses the investigation of mechanisms of gene-environment interplay driving social inequalities across developmental trajectories from childhood to early adulthood. Genetic variation and social experiences are relevant sources of individual differences in life outcomes and contribute to the development and the transmission of social inequalities (Spinath & Bleidorn, 2017). Genes and experiences of environments reciprocally interact with psychological and social factors which are relevant for inequality, such as personality traits, available resources, educational and career paths, or social and political integration. In this context, research using TwinLife data is often driven by the assumption that the expression of one's genetic potential depends on environmental circumstances (Scarr, 1992, 1993), with synergistic and dynamic interplays between genetic predispositions and environments (Kandler et al., 2021). First, the interplay can be described by gene-environment *interactions*, with mechanisms of the social context to a) trigger or b) compensate a genetic predisposition, c) enhancing the realisation of genetic potential or d) acting as a social control for behaviour when there is a genetic predisposition (Shanahan & Hofer, 2005). Second, a selective exposure of genetic predisposition to different environments is described as gene-environment *correlation*. This can be passive (e.g., parents create life circumstances that fit to their own and thus to their children's predispositions), active (i.e., individuals actively seek, create, or manipulate these circumstances), or reactive (i.e., individuals evoke certain circumstances) (Plomin et al., 1977). The active or reactive kinds occur, for example, during adolescence, when individuals increasingly shape and more autonomously select, avoid, or transform their environment according to their genetic predispositions (Scarr & McCartney, 1983). The investigation of genetic and environmental sources of variation, their correlation and interaction regarding educational inequality, is at the heart of the TwinLife study. It allows a better understanding of the underlying processes leading to differences in life outcomes (Diewald et al., 2015; Hahn et al., 2016).

One core assumption that elucidates the importance for genetically informative analysis of social stratification and inequalities is known as the Scarr-Rowe hypothesis, proposing a moderating effect of low socio-economic status and social disadvantage in childhood limiting the unfolding of genetic potentials for intelligence (Rowe et al., 1999; Scarr-Salapatek, 1971). This effect on intelligence and with extension to educational attainment has been investigated using TwinLife data (Baier & Lang, 2019; Gottschling et al., 2019). Findings on cognitive abilities and educational outcomes based on TwinLife data are summarised by Mönkediek et al. (2019)

and Lang et al. (2019). Further studies on educational phenomena focus on heritability and environmental effects on numeracy and literacy (as measured by school grades in mathematics and German; Eifler et al., 2019), effects of parental separation (Baier & Van Winkle, 2021), or on a trait-specific investigation of the equal-environment assumption, i.e., the assumption that mono- and dizygotic twins share environmental influences to an equal extent (Mönkediek, 2021). Educational predictors and outcomes have been analysed from numerous perspectives, including influences of unequal socio-economic backgrounds (Mönkediek & Diewald, 2021; Paulus et al., 2021; Stienstra et al., 2021), parenting styles (Grätz et al., 2022), and motivational and personality variables (Dings & Spinath, 2021). Other investigations focus on the specifics of environmental influences on educational attainment (Eifler & Riemann, 2022; Starr & Riemann, 2022).

TwinLife includes longitudinal data from monozygotic and dizygotic same-sex twin pairs and their families within four age cohorts (5, 11, 17 and 23 years), aiming to investigate biological, psychological, and social origins of social inequality in six broad areas: (1) skill formation and education, (2) career and labour market attainment, (3) social integration and participation, (4) subjective perception of quality of life, (5) physical and psychological health, and (6) deviant behaviour and behavioural problems. TwinLife is the first German twin study using a population register-based sampling design (Hahn et al., 2016; Lang et al., 2019; Mönkediek et al., 2019), containing $N = 4,096$ families at first data collection in 2014. Respondents are surveyed annually, alternating face-to-face interviews in the household with telephone interviews.

TwinLife follows a multi-method approach to study the education and qualification histories of participants, presenting objective measures on school achievements and parental reports on children's academic performance. As objective measures, TwinLife data include the most recent school report cards transferred into a generalised scheme (Instinske et al., 2022). Furthermore, as a proxy for general cognitive ability, results from assessments of the Culture Fair Test (CFT; Weiß, 2006; Weiß & Osterland, 2012) are available. For children under the age of ten, the assessment was applied in a paper-and-pencil version administered by a trained interviewer, whereas older participants completed a computer-based version of the CFT (Klatzka et al., 2022). As further measures related to education, child and parental reports on motivation, personality, school and home characteristics are collected. Examples include academic self-concept, learning and achievement motivation, competence ratings on skills, self-esteem, as well as student-teacher-interaction, pressure and stress at school, special educational treatment, perceived chaos at home, as well as frequency and duration of media use. In addition,

TwinLife contains information on educational and career aspirations and assessments of the probability of achieving set targets. The study covers an age range of all major educational transitions, from primary to secondary to tertiary education, until first years of professional career. The measured constructs not only cover information on educational transitions but also further life events (e.g., moving out of parental home, starting a relationship or family) that can be related to educational and career decisions.

TwinLife offers rich data on within- and outside-familial environments. Within-family characteristics comprise, among others, socioeconomic status, family structure, home environment, and parenting. Outside-family characteristics encompass, for example, neighbourhood, schooling, and social networks. The wide range of characteristics describing within- and outside-familial environments provide excellent conditions for the longitudinal investigation of educational processes, including analyses regarding skill formation, and the relevance of intra- and extra-familial socialisation processes. Beyond that, the study addresses the consequences of inequalities in educational processes for differences in labour market attainment, social integration, and subjective quality of life.

## 2 METHODS

### 2.1 STUDY DESIGN

TwinLife is a longitudinal, cross-sequential study with an extended twin family design, including data from twins and their parents, as well as their non-twin siblings, partners, and children, if available. The sample consists of four age cohorts of monozygotic and dizygotic same-sex twin pairs who were 5, 11, 17, and 23 years old at the time of the first survey. Each age cohort includes two birth years (subsamples A and B) who are interviewed in consecutive years. Further information on the cross-sequential survey design is presented in the TwinLife short guide (Krell et al., 2021). With regard to educational trajectories, the youngest twins (cohort 1) were first interviewed before school entry at age 5 and the oldest twins (cohort 4) were first interviewed at age 23, being within tertiary education or having completed vocational training and beginning to establish themselves on the labour market.

### 2.2 TIME OF DATA COLLECTION

The first data collection took place in the form of a household interview (Face-to-Face 1 = F2F 1) from October 2014 to April 2016. Approximately one year after respondents participated in the F2F 1 data collection, the computer-assisted telephone interview (CATI 1) was conducted. This data collection lasted from November 2015 to April 2017. Subsequent data collections (F2F 2 in 2016–2018, CATI 2 in 2017–2019, F2F 3 in 2018–2020,

CATI 3 in 2019–2021, and F2F 4 in 2020–2022) were also conducted at this rate. The time interval between two data collections is always one year, and participants are interviewed alternately at home and by telephone. During the corona-virus pandemic, interviews of the F2F 4 data collection were conducted by telephone. The CATI 4 data collection began in December 2021 and is expected to be completed in spring 2023. The most recent and final data collection, F2F 5, started in fall 2022 and is expected to be concluded in spring 2024. The time frame of data collections is presented in Figure 1.

Furthermore, three COVID-19 supplementary surveys were conducted, as depicted in Figure 2. The first supplemental COVID-19 survey (Cov 1) retrospectively assessed behaviour, attitudes, health, stresses and socio-economic changes from the start of a strict lockdown in March 2020 until the first relaxation of pandemic containment measures and was conducted from July to November 2020. The second supplemental COVID-19 survey (Cov 2) was part of the F2F 4 data collection and CATI 3 data collection and respondents were surveyed from December 2020 to June 2021. A third supplemental COVID-19 survey (Cov 3) was conducted in autumn 2021 and within the data collection at F2F 5, which started in autumn 2022, a fourth COVID-19 supplementary survey was implemented.

In addition, please note that two saliva samples were collected from TwinLife participants (see Figure 2) via a saliva self-collection kit (Oragene kit, 2 ml) with the aim to genotype respondents and to identify changes in DNA methylation based on epigenetic mechanisms. A third saliva sample is collected along with the final data collection (F2F 5). For the first and second saliva sample, data of roughly 5,500 and 3,300 participants are available, respectively.

### 2.3 LOCATION OF DATA COLLECTION

The survey takes place throughout Germany with rural and urban regions.

### 2.4 SAMPLING, SAMPLE AND DATA COLLECTION

The data collection was carried out only in Germany and limited to families with good German language proficiency. Due to the lack of a twin registry in Germany, a national probability-based sampling procedure was carried out. First, a random sample of 500 communities was drawn from a total of approximately 11,900 communities. Three subsamples of communities were then selected: a "basic sample" (180 municipalities with >= 10,000 inhabitants), an "urban sample" (60 municipalities with >= 50,000 inhabitants), and a "rural sample" (260 municipalities with >= 5,000 to <20,000 inhabitants). For more information on this threefold random sampling approach, see Brix et al. (2017). To identify twin families, these communities were screened

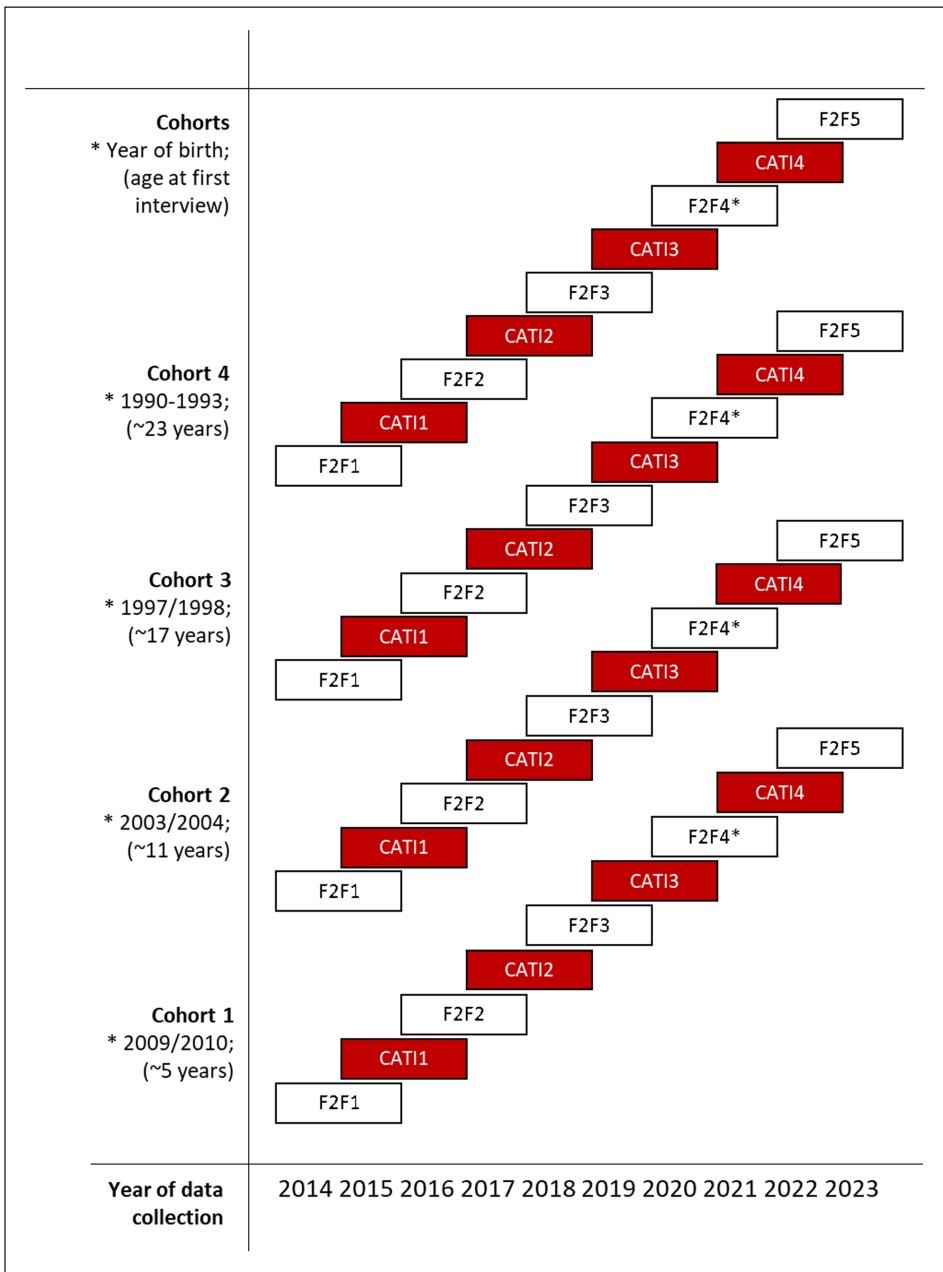

**Figure 1** Data Collections in TwinLife.

*Note*: F2F: Face-to-face interview at household; CATI: Computer-assisted telephone interview, *The fourth face-to-face interview in the household had to be substituted with a CATI & CAWI (computer assisted web interview) due to the COVID-19 pandemic, in the fifth face-to-face data collections, participants can choose between a conventional household interview and a CATI & CAWI variant of the survey.

for persons of the same sex with the same date of birth who were registered at the same address according to the current population register. For the oldest age cohort, earlier addresses also had to be considered. This resulted in a gross sample of more than 13,000 families. Finally, $N$ = 4,097 families with monozygotic or dizygotic same-sex twin pairs participated in the first TwinLife survey (F2F 1; for more detailed information, please see Lang & Kottwitz, 2020). However, one family was removed from data release in version v4-0-0 (https://doi.org/10.4232/1.13539) due to unresolvable inconsistencies in the family's responses, resulting in a total number of $N$ = 4,096 families in the dataset.

The TwinLife sample comprises four age cohorts with twins aged 5 (cohort 1), 11 (cohort 2), 17 (cohort 3), and 23 (cohort 4) at the time of the first survey and consists of two subsamples (A and B in Figure 2). This is due to the initial sampling of the age cohorts to achieve a sufficiently large sample size (Krell et al., 2021). In each age cohort, families with twins born in two consecutive years are included in the sample. The first birth cohort of an age cohort forms subsample A, and the second birth cohort forms subsample B. These subsamples are surveyed in consecutive years so that twins are interviewed at the same age. In total, each age cohort includes approximately 1,000 families. Not only the

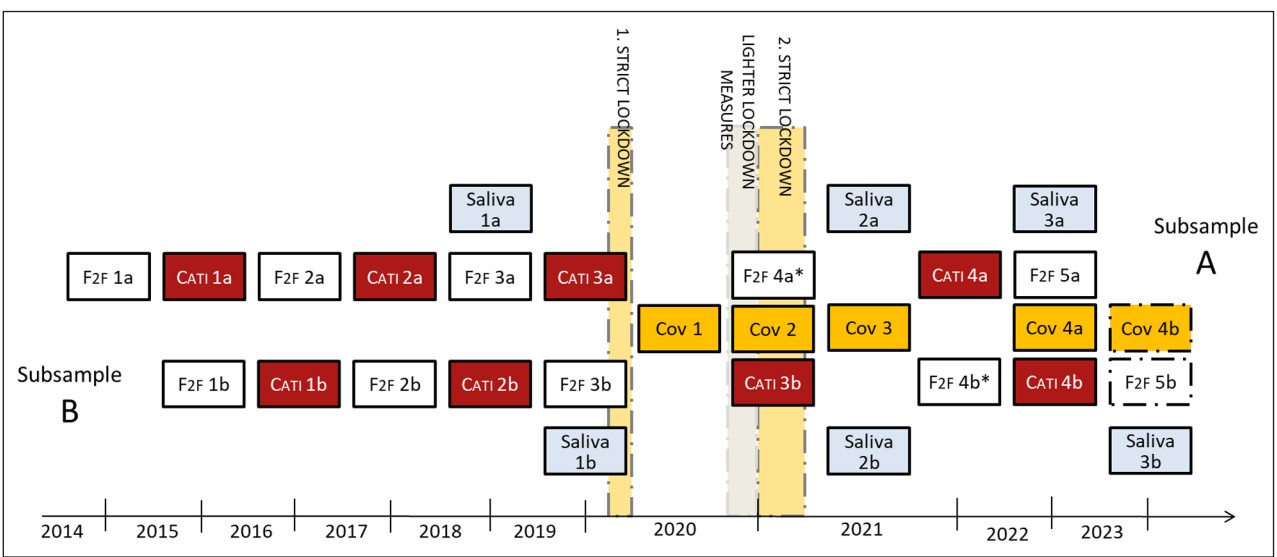

**Figure 2** TwinLife COVID-19 Supplementary Surveys.

*Note*: F2F 1a – F2F 5b: First face-to-face interview for subsample A – Fifth face-to-face interview for subsample B; CATI 1a – CATI 4b: First computer-assisted telephone interview for subsample A – fourth computer-assisted telephone interview for subsample B; Saliva: collection of saliva via self-collection kit (Oragene kit, 2 ml); Cov 1 – Cov 4b: First COVID-19 supplementary survey – Fourth COVID-19 supplementary survey for subsample B; *The fourth face-to-face interview in the household had to be substituted with a CATI & CAWI (computer assisted web interview) due to the COVID-19 pandemic. In the fifth face-to-face data collection, participants can choose between a conventional household interview and a CATI & CAWI variant of the survey; 1. strict lockdown and 2. strict lockdown: contact restrictions in private life, travel restrictions, closure of schools, gastronomy, retail, service sector, cultural institutions (e.g., museums, cinema, concerts) and social sports activities; lighter lockdown measures: contact restrictions in private life and closure of gastronomy, cultural institutions and social sports activities.

twins themselves are interviewed, but also their parents (biological and step-parents, if applicable), a sibling aged 5 years or older (if there is more than one sibling, the sibling closest in age to the twins is considered), and, if the twins are 18 years old or older, the twins' current partner. Siblings and partners are interviewed only if they live with one of the twins. This procedure led to an initial sample size of roughly 16,500 persons being interviewed. From F2F 4 onwards, data on children of adult twins are also collected, if applicable.

The development of the sample size is shown in Figure 3. TwinLife experienced a stronger decline in participation rates after the first wave of the survey, which is typical for panel studies. However, participation rates stabilised over the course of the study for all cohorts and have been comparable to other panel studies so far (e.g., pairfam: Brüderl et al., 2022; SOEP: Siegers et al., 2022).

Analyses of the first wave of the survey have shown that many characteristics of TwinLife households are comparable to those of multi-child households in the Microcensus (Lang & Kottwitz, 2020). Also, selective drop-out in terms of personality traits and relationship characteristics cannot be observed in TwinLife (Klatzka et al., 2019). Nevertheless, as in other panel studies, non-response or panel dropouts may bias estimates if respondents differ systematically from non-respondents, or if dropouts differ from those who remain. Table 1 shows the distribution by zygosity of the twins, sex, migration background, and education of the twins' mother across all data collections. Cohort specific distributions of these

characteristics across all data collections are presented in the Appendix (Figure A1 to A4). There is no trend toward overrepresentation of monozygotic twins or of any sex. At all survey time points, approximately 55% of participating twins are female and 45% are monozygotic. However, families with low-educated mothers or with a migration background were slightly more likely to drop out of the panel after the initial data collections. The dropout of respondents with a migration background can be partly explained by the fact that the questionnaires are only available in German and, due to their complexity, are only suitable for families with a good knowledge of German. To account for selective non-response, as well as panel attrition in general, we created nonresponse and panel weights. In addition, the TwinLife sample design described above is a stratified random sampling design in which families from certain regions have a higher probability of being sampled than families from other regions. The design weight addresses unequal sampling probabilities introduced by this design. These panel weights have been available since data release 6-0-0. For more information on fieldwork outcomes, see the methodology reports for the individual data collections (accessible via https://www.twin-life.de/documentation/downloads).

In the F2F surveys, target respondents are interviewed in their households. In the CATI surveys, and during the Corona pandemic in the F2F 4 survey, respondents are interviewed by telephone. From the CATI 3 survey onward, a computer assisted web

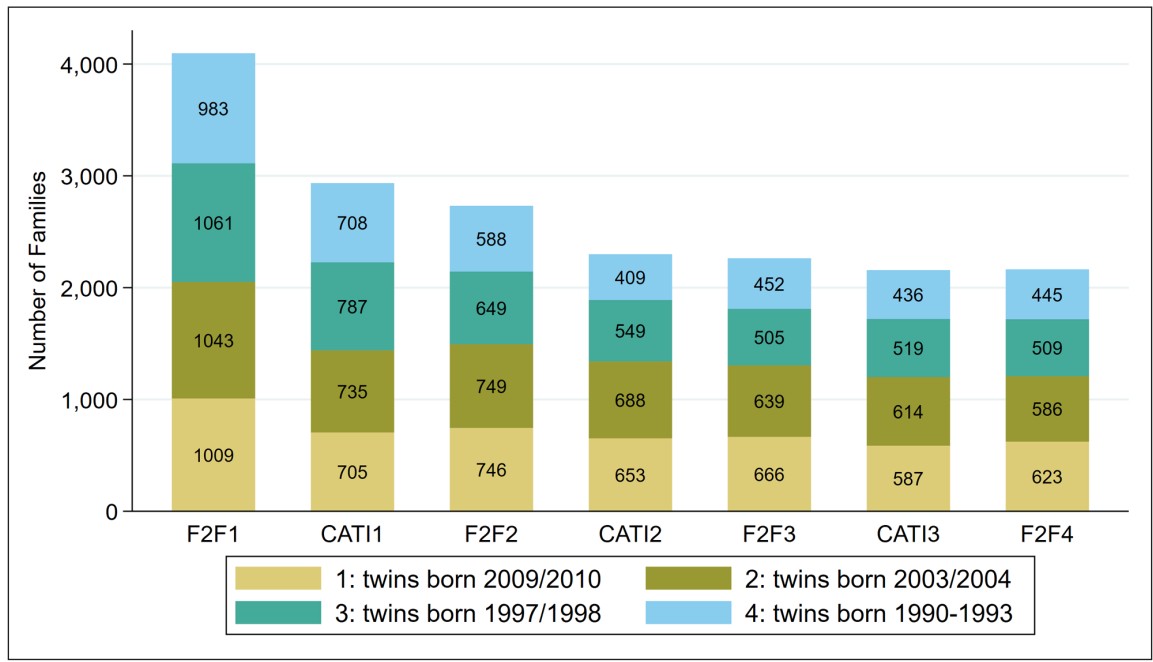

**Figure 3** Sample size development.

*Note*: Calculations are based on TwinLife Data file Version 6.1.0, https://doi.org/10.4232/1.13987 (F2F1 to F2F3) and preliminary data for recently collected data (CATI3, F2F4).

|  | DATA COLLECTIONS | | | | | | |
|---|---|---|---|---|---|---|---|
|  | **F2F 1** | **CATI 1** | **F2F 2** | **CATI 2** | **F2F 3** | **CATI 3** | **F2F 4** |
| **Zygosity** | | | | | | | |
| Monozygotic | 1880 (46%) | 1332 (45%) | 1240 (45%) | 1050 (46%) | 1019 (45%) | 970 (45%) | 968 (45%) |
| Dizygotic | 2213 (54%) | 1603 (55%) | 1491 (55%) | 1249 (54%) | 1243 (55%) | 1185 (55%) | 1195 (55%) |
| Total | 4093 (100%) | 2935 (100%) | 2731 (100%) | 2299 (100%) | 2262 (100%) | 2155 (100%) | 2163 (100%) |
| **Sex of twins** | | | | | | | |
| Male | 1854 (45%) | 1330 (45%) | 1257 (46%) | 1051 (46%) | 1037 (46%) | 977 (45%) | 992 (46%) |
| Female | 2242 (55%) | 1605 (55%) | 1475 (54%) | 1248 (54%) | 1225 (54%) | 1179 (55%) | 1171 (54%) |
| Total | 4096 (100%) | 2935 (100%) | 2732 (100%) | 2299 (100%) | 2262 (100%) | 2156 (100%) | 2163 (100%) |
| **Migration background of twins** | | | | | | | |
| No migration background | 3074 (75%) | 2296 (78%) | 2155 (79%) | 1806 (79%) | 1799 (80%) | 1725 (80%) | 1734 (80%) |
| Migration background | 1022 (25%) | 639 (22%) | 577 (21%) | 493 (21%) | 463 (20%) | 431 (20%) | 429 (20%) |
| Total | 4096 (100%) | 2935 (100%) | 2732 (100%) | 2299 (100%) | 2262 (100%) | 2156 (100%) | 2163 (100%) |
| **Mother's highest education at F2F1** | | | | | | | |
| Primary level (1) | 730 (18%) | 405 (14%) | 335 (12%) | 251 (11%) | 242 (11%) | 201 (9%) | 219 (10%) |
| Intermediate level (2) | 2031 (50%) | 1463 (50%) | 1328 (49%) | 1096 (48%) | 1090 (48%) | 1039 (48%) | 1039 (48%) |
| Higher level (3) | 1327 (32%) | 1065 (36%) | 1066 (39%) | 950 (41%) | 927 (41%) | 914 (42%) | 901 (42%) |
| Total | 4088 (100%) | 2933 (100%) | 2729 (100%) | 2297 (100%) | 2259 (100%) | 2154 (100%) | 2159 (100%) |

**Table 1** Sample characteristics.

*Note*: Calculations are based on TwinLife Data file Version 6.1.0, https://doi.org/10.4232/1.13987 (F2F1 to F2F3) and preliminary data for recently collected data (CATI3, F2F4). Migration background is assigned if either the twins were born abroad (first generation) or at least one parent was born abroad (second generation). Missing self-reports of the parents are filled in with the twins' proxy information on the parents' place of birth. In the case of completely missing information, it is assumed that there is no migration background. Mother's education at F2F1 is based on CASMIN; missing information has been filled with father's education.

interview (CAWI) is conducted when adult respondents cannot be reached by telephone or if respondents want to use that option instead of a telephone interview. In contrast to the F2F surveys, children have to be at least 10 years old to be interviewed themselves in the CATI surveys. The interview begins with a family questionnaire answered by one person, either a parent or a twin aged 18 or older, to identify the individuals to be interviewed. A household questionnaire is then completed in each household with at least one target person. For example, in the youngest age cohort, twins, parents and siblings often live in the same household, whereas in the oldest age cohort, it occurs that twins and parents live in different households. To keep the duration of the F2F survey in the households as short as possible, multiple instruments are used so that interviews can be conducted simultaneously. These include computer-assisted face-to-face interviews conducted by an interviewer, computer-assisted self-interviews via tablet computers, and paper-and-pencil interviews. For F2F interviews and CATI interviews after CATI 1, the respondents are paid up to 10 Euro per person (except, families of cohort 1 and 2 who are paid 20 Euro for participation in CATI 2), or a gift was given to respondents in the F2F 1 survey instead of payment.

In the COVID-19 supplementary surveys, diverse interview methods were used. While the first COVID-19 supplementary questionnaire was conducted online (with approximately 4,300 persons participating), the second and third supplementary survey used an online-telephone mixed format or a paper-pencil-online mixed format. In the second and third supplementary questionnaire, roughly 9,600 and 4,000 participants took part, respectively. While no additional monetary incentive was given for the first or second supplementary survey, the participation in the third supplementary survey was rewarded with a pre-incentive of 5 Euro per person.

## 2.5 MATERIALS/SURVEY INSTRUMENTS
TwinLife uses a wide variety of survey instruments. A complete overview and detailed information on the wording and adaptive study item administration (e.g., filtering of study items based on age) can be gained via the TwinLife documentation website (https://www.twin-life.de/documentation/). Information on scientific scales adapted for the TwinLife study can be found in the scales manual (Klatzka et al., 2022). As a complete overview of all items and scales would be too extensive, we focus on presenting core aspects relevant for measuring or predicting educational success for participants still enrolled in school in Table 2. For more information, please consult our documentation website (https://www.twin-life.de/documentation/). Please note that the constructs depicted in Table 2 are not necessarily assessed in every data collection of TwinLife.

## 2.6 QUALITY CONTROL
A professional survey institute (TNS Infratest at F2F 1; infas Institute for Applied Social Sciences from CATI 1 onward) is commissioned with conducting the interviews and uses experienced and trained interviewers for fieldwork. In the beginning of the project, several pretests were conducted in order to secure the practicability of the survey. To guarantee the quality of data, several checks are carried out by the TwinLife team as well as the survey institute. Data of participants is thoroughly checked to ensure that: a) basic information (person type, sex, names and age) are consistent longitudinally, b) no duplicate information is present, and c) filtering conditions for questions are correct. If possible, data errors or implausible information are corrected. For selected constructs, data is also checked for content validity (e.g., cognitive abilities, see Gottschling, 2017; height and weight data, see Klatzka et al., 2021) and errors are corrected in generated variables.

## 2.7 DATA ANONYMISATION AND ETHICAL ISSUES
Ethical approval for the TwinLife study was received from the German Psychological Association (Deutsche Gesellschaft für Psychologie, protocol numbers: RR 11.2009 and RR 09.2013). Ethical approval for the first supplemental COVID-19 survey was received from the ethics committee at the University Bielefeld (07.2020). All data from study participants are collected by an external survey institute and pseudonymised through assignment of personal and family ID numbers for each individual.

Prior to each data collection, participants received a data protection notice informing them about the processing and use of their personal data in accordance with the data protection law and voluntariness of study participation.

## 2.8 EXISTING USE OF DATA
Publications related to education using TwinLife data are listed in the Appendix. A complete overview of all publications based on TwinLife data including other subject areas can be found at https://www.twin-life.de/publikationen.

# 3 DATASET DESCRIPTION AND ACCESS
## 3.1 REPOSITORY LOCATION
The data can be accessed via the GESIS data catalogue (https://search.gesis.org/research_data/ZA6701).

## 3.2 OBJECT/FILE NAME
The repository contains files of four classes: 1) The datasets, 2) the questionnaires, 3) the codebooks, and 4) other documents (e.g., the scales manual). Data from the most recent available version of the TwinLife study,

| CONSTRUCT | DESCRIPTION | EXAMPLE ITEMS |
|---|---|---|
| **Educational attainment** | | |
| Information on education history and educational institutions | TwinLife assesses extensive information on education history and educational institutions including school attendance, type of school, grade level, skipping or repeating grade levels, recommendation for secondary school, desired school-leaving qualifications, after school-jobs, information on vocational training, and higher education | "Are you currently in education or training? In other words, are you attending school or higher education (including doctorate/Ph.D.), completing an apprenticeship or vocational training, or in further education or training?" |
| Tutoring and extracurricular activities | It is assessed whether children take part in extracurricular activities, including help with homework, remedial teaching, additional classes, and participation in clubs | "Does your child participate outside regular school hours in one or more of the following activities?" – Help with homework |
| Information on school grades and school report cards | Photographs of the report card were taken and coded by trained personnel (for more information, see Instinske et al., 2022). If there was no certificate available or the participants did not give their consent, questions about the school grades in maths and German were asked instead. For participants that have left school already, the grade point average was surveyed. | "Please indicate what school grade you had on your last report card in the following subjects"- Math/ German |
| Cognitive abilities | The Culture Fair Test (CFT; Weiß, 2006; Weiß & Osterland, 2012) was applied. The test measures non-verbal (fluid) intelligence, which can serve as a proxy for general cognitive ability. For participants aged five to nine years, three different subtests (figural reasoning, figural classification, and matrices; CFT 1-R; Weiß & Osterland, 2012) were assessed. For participants ten years of age and older, four subtests were used (reasoning in addition to the aforementioned three subtests; CFT 20-R; Weiß, 2006). For more information, see Gottschling (2017). | See test manual of the CFT |
| **Motivation and academic self-concept** | | |
| Academic self-concept | Academic self-concept was assessed for school students with three items concerning their academic self-concept for school in general, but also three items specific for the subject maths and German. The assessment for school students was based on scales for academic self-concept (Dickhäuser et al., 2002; orig. "Skalen zum akademischen Selbstkonzept" (SESSKO)). | "I am [not talented – talented] for school" <br> "I know [Just a little – A lot] in maths" |
| Intrinsic motivation | Intrinsic motivation was assessed for school students with three items concerning their intrinsic motivation for school in general, but also three items specific for the subject maths and German. Intrinsic motivation for school students was measured with adapted items of the scale for the assessment of subjective school values (Steinmayr & Spinath, 2010; orig. "Skala zur Erfassung subjektiver schulischer Werte" (SESSW)). | "I like doing the things I learn at school." <br> "Maths is fun." |
| Learning motivation | The scales for the assessment of learning and performance motivation (Spinath et al., 2002); orig. "Skalen zur Erfassung der Lern- und Leistungsmotivation" (SELLMO-S) were adapted to measure learning motivation. The construct was assessed for school in general using three items. | "At school, I am interested in learning something interesting." |
| Achievement motivation | Three items were developed for TwinLife. | "Good achievements mean a lot to me." |
| **Subjective perceptions of burden due to school and perception of the school climate** | | |
| Feelings of burden due to school | Feelings of burden were assessed with an adapted version of questions originating from a survey by the German National Educational Panel Study (NEPS, 2012), called "additional study in Thuringia" (Blossfeld & Roßbach, 2019). It was assessed for all school students aged 13 or older with seven items. | "I often feel tense when I come home from school." |
| Student-teacher interactions | This construct was assessed with five items, originating from the PISA survey (OECD, 2013). | "Students get along well with most teachers." |

(Contd.)

| CONSTRUCT | DESCRIPTION | EXAMPLE ITEMS |
|---|---|---|
| **Personality and other traits** | | |
| Personality | Two different versions of the Big Five Inventory were used. Every participant over ten years of age rated their own personality on the Big Five Inventory – Short Version (BFI-S; Gerlitz & Schupp, 2005). The sub-scales conscientiousness, extraversion, agreeableness and neuroticism were assessed with three items each, while openness was assessed with four items.<br>For younger children, their parents rated their personality on the Five factor questionnaire for children – short form (FFFK-K; Weinert et al., 2007); orig. "Fünf Faktoren Fragebogen für Kinder – Kurzform", with 2 items for each subscale. | "I see myself as someone who does a thorough job."<br>"I see myself as someone who is talkative."<br>"My child is [Not that interested – Hungry for knowledge]."<br>"My child is [Untidy – Tidy]." |
| Self-Regulation | Self-Regulation was assessed using items of two different questionnaires: Three items of the BISS scale, which is a German adaption of the Grit Scale (Consistency of Interests) from Fleckenstein et al., (2014); and three items of the German short version of the Self-Control Scale (SCS-K-D); Bertrams & Dickhäuser, (2009). For children aged nine or younger, only the Self-Control Scale was assessed via parental report on their children. | "New ideas and projects sometimes distract me from previous ones."<br>"I do certain things that are bad for me, if they are fun." |
| Self-Eesteem | Depending on the age of children, self-esteem was assessed either as a self-report or as a parental report. As a self-report, self-esteem was assessed using three items from the „Panel Analysis of Intimate Relationships and Family Dynamics" (pairfam; see Thoennissen et al., 2014; items are based on the Rosenberg Self-Esteem Scale; RSE; Rosenberg, 1965). For the parental report (participants aged 5 to 12), two of these items were rephrased. | "I take a positive attitude toward myself."<br>"My child is self-confident. |
| Self-Efficacy | Three items from the general self-efficacy short scale (ASKU; Beierlein et al., 2012; orig. "Allgemeine Selbstwirksamkeit Kurzskala") were used to measure this construct. | "I can rely on my own abilities in difficult situations." |
| **Home environment and parental support** | | |
| Quality of home environment | This construct was measured with an adapted version of the confusion, hubbub, and order scale (CHAOS; Matheny et al., 1995). It can be used to rate chaotic vs. orderly home environments. The scale consisted of six items. | "You can't hear yourself think in our home." |
| Parental involvement | The scales for parental involvement were inspired by the CoSMoS project (see Spinath & Wolf, 2006) and an instrument by Lorenz & Wild (2007). The items were assessed as a child report on their parents for four different facets with three items each: Structure, emotional support, autonomy and control. | "When I study for an exam, I know exactly how much effort my parents expect of me."<br>"When I get a poor grade, my parents complain and demand that I work harder." |
| Parenting style | The scales for parenting style were inspired by pairfam (Huinink et al., 2011). Parenting style was assessed as parental self-report of each parent and child report on both their parents. There were 5 scales: Emotional warmth (four items), psychological control (three items), negative communication (two items), monitoring (two items), inconsistent parenting (two items) | "You show with words and gestures that you like your child."<br>"If your child does something against your will, you punish your child."<br>"Your mother/your father yells at you because you did something wrong." |
| Family activities | Parents and children were asked how much time they spent together doing certain activities, like reading, singing or going to the playground in the last month. The questions were taken from the pairfam study. | "How often have your parents or other members of your family taken part in the following activities with you during the last four weeks?" – Singing and making music |
| Cultural capital | Depending on the age of the participant, cultural capital was assessed either as a self-report or as a parental report. The self-report items originated from the NEPS study (for more information regarding particular subscales, see Goßmann, 2018). Items for cultural capital included the categories embodied cultural capital (five items), cultural involvement (five items), and participation in high culture (five items). As a parental report, however, only participation in high culture was asked, with similar questions as in the self-report. Also, the participants were asked how much time they spent reading daily. | "Do you have books of poems at home"<br>"In general, how often do you discuss the following things with others? – Political and social issues" |

(Contd.)

| CONSTRUCT | DESCRIPTION | EXAMPLE ITEMS |
|---|---|---|
| Social relationships | We collected data on the number of friends, age and sex of significant others, and also on the possible experience of loneliness.<br>In addition, information was collected on sibling and parent-child relationships, support relationships, and information on conflicts and quarrels with others. | "How many close friends do you have? This includes family members who you are close to. This does not include Facebook friends."<br>"To what extent do you agree with the following statement? I often feel lonely."<br>"How satisfied are you with your relationship with your twin/sibling?" |
| COVID-19 pandemic | | |
| Pandemic related topics | In light of the COVID-19 pandemic, the TwinLife project decided to incorporate a special questionnaire series with COVID-19 pandemic specific questions to gain information on pandemic related changes in some of the core social inequality dimensions investigated in TwinLife: Burden and threats due to the COVID-19 pandemic; emotional stress and COVID-19 related coping and resilience; changes in employment and finances; worsening of pre-existing medical condition; changes in educational opportunities. | "During the Corona pandemic, schools were or are closed for extended periods of time. Are you attending classes normally right now?"<br>"The Corona pandemic continues to affect the daily lives of many people. If you are currently experiencing the following limitations, how bad are they for you?" – Having schooling (also) at home |

**Table 2** Overview of constructs in TwinLife with relevance for educational research.

release 6.1.0 is delivered in the following files (listed are download files in Stata, the SPSS files are organised in the same way):

- ZA6701_family_wide_wid1_v6-1-0.dta
- ZA6701_family_wide_wid2_v6-1-0.dta
- ZA6701_family_wide_wid3_v6-1-0.dta
- ZA6701_family_wide_wid4_v6-1-0.dta
- ZA6701_family_wide_wid5_v6-1-0.dta
- ZA6701_master_v6-1-0.dta
- ZA6701_mode_wid1_v6-1-0.dta
- ZA6701_person_cov1_v6-1-0.dta
- ZA6701_person_cov2_v6-1-0.dta
- ZA6701_person_unadj_wid1_v6-1-0.dta
- ZA6701_person_unadj_wid3_v6-1-0.dta
- ZA6701_person_unadj_wid5_v6-1-0.dta
- ZA6701_person_wid1_v6-1-0.dta
- ZA6701_person_wid2_v6-1-0.dta
- ZA6701_person_wid3_v6-1-0.dta
- ZA6701_person_wid4_v6-1-0.dta
- ZA6701_person_wid5_v6-1-0.dta
- ZA6701_weights_v6-1-0.dta
- ZA6701_zygosity_v6-1-0.dta

Further information on data formats and the content of the data files is presented in the TwinLife short guide (Krell et al., 2021).

### 3.3 DATA TYPE
The TwinLife datasets contain primary, generated and processed data.

### 3.4 FORMAT NAMES AND VERSIONS
Datasets are delivered in the following formats: .dta (Stata version 13), .sav (SPSS version 28). Data documentation uses different formats, such as Excel files (.xlsx) and pdf files (.pdf).

### 3.5 LANGUAGE
The short-guide, codebooks, a longitudinal overview of assessed constructs, the scales manual, methodology reports, the documentation of generated variables and technical reports are given in English language. Original questionnaires are in German language and the datasets include both English and German labels.

### 3.6 LICENCE
The data are openly accessible for academic research and teaching after written permission from the GESIS Data Archive. For this purpose, a data use contract is concluded, in which data recipients agree to follow legal and organisational requirements to ensure data confidentiality. There are no restrictions based on research topics, however it is ensured that the data is used for scientific purposes only. Every researcher, no matter her or his affiliation or experience level, will receive data access after signing the data usage agreement. Furthermore, there are no restrictions to data access for international researchers (within and outside the EU).

### 3.7 LIMITS TO SHARING
The TwinLife data set is made available as a scientific use file only after signing a data usage agreement.

### 3.8 PUBLICATION DATE
New data collections are published once a year. The data (all data releases up to Face-to Face 3) was last published on 07 September 2022 (DOI: https://doi.org/10.4232/1.13987).

### 3.9 FAIR DATA/CODEBOOK
The TwinLife data follow the FAIR Guiding Principles (Wilkinson et al., 2016). The documentation for all datasets can be found on the study website (https://www.

twin-life.de/documentation/downloads) or accessed via GESIS (https://search.gesis.org/research_data/ZA6701). For detailed information on variables and instruments, see https://paneldata.org/twinlife/. Most recent available version of the study: 6.1.0, https://doi.org/10.4232/1.13987.

## 4 REUSE POTENTIAL

Due to its large scope of diverse measurements, the TwinLife data offer a great potential for reuse, allowing research in many different areas and application of non-genetic as well as behavioural genetic analysis methods.

First, in addition to the possibility of exploring the development of educational inequalities and their consequences, the data are suitable for researching the emergence of social inequalities in various domains of life and for investigating genetic and environmental influences and their interplay. Using the information available in TwinLife, inequalities in six domains of life can be addressed. Besides the domain of (1) skill formation and education, these domains include: (2) career and labour market attainment, (3) political and social integration and participation, (4) subjective perception of quality of life, (5) physical and psychological health, and (6) deviant behaviour and behavioural problems (Mönkediek et al., 2019). Additionally, beginning in 2020, a questionnaire was implemented on the COVID-19 pandemic and associated changes and burdens on respondents' lives in various areas, including education (home schooling, student support, etc.).

Second, the majority of constructs included in TwinLife were taken from or adapted from other large German surveys, including the Socio-Economic Panel (SOEP), the German Family Panel (pairfam), and the National Educational Panel Study (NEPS). This enables combined analyses (i.e., pooling of data) and comparisons of results between twin and non-twin populations. The TwinLife data therefore offer opportunities for meta- and mega-analyses.

Third, the extended twin family design not only allows the analyses of individuals or twin pairs, but also other dyads (parent-child and spouses) and the analysis of family groups in the context of a multi-actor design. When looking at differences across siblings, for example, the data provides opportunities to examine the relevance of within-family inequalities, as well as differences in the effects of sibling- and family-specific environments. Furthermore, by comparing indicators between parents and their children, the data allow the examination of processes related to the intergenerational transmission of social inequalities. Looking at the parents of twins or the twins' partners, it is possible to study processes of partnership formation and to address questions regarding the relevance of educational homogamy

for the course of partnerships and its implications for the intergenerational transmission of educational inequalities.

Fourth, the cross-sequential design of the study offers the advantage that the importance of influencing factors for the development and consequences of social inequalities can be considered and studied from a life course perspective. From the F2F 4 data collection onward, the included cohorts overlap in age (e.g., cohort 1 in data collection F2F 4 has the same age as cohort 2 in data collection F2F 1) so that the data can also be pooled for analyses to disentangle age, time, and cohort effects. Thus, the reported data cover a large part of the life span of individuals during which the most important foundations for later life chances are laid, such as their educational career, the entrance into the labour market, and the phase of family formation. Taken together, this supports theory building around questions about the longer-term impact of genetic and environmental influences, and their interplay, for inequalities that existed early on. Thereby it is possible to study the accumulation of disadvantage over the life course, as well as possible compensatory or accentuating mechanisms (e.g., Mönkediek & Diewald 2021).

Fifth, the broad range of information collected offers unique opportunities for researchers from different disciplines to collaborate on common research topics. The fact that the sample was additionally genotyped based on saliva samples in the F2F 3 data collection of TwinLife further increases the reuse potential of the data due to its opportunity to combine a twin design with molecular genetic analysis in the near future. For an overview on the advantages of molecular genetic information in twin samples, see Harden (2021).

A limitation of the reported data is that in many cases short versions of established scales had to be used to reduce the time interviewers spent in the households. In addition, data on fathers is partly missing because they were more difficult to reach than the mothers of the twins. Similarly, the older cohorts of twins were more difficult to reach in the later data collections, reducing the sample size in the later data collections to some extent. Finally, the reported data only include same-sex twin pairs, which limits the possibilities to study sex-specific effects.

Overall, however, the TwinLife data show a high potential for reuse. The wide thematic range of the study, as well as the expansion of the survey data by including genetic data, can make an important contribution to theory building by further investigating the interplay of genes and environment for the genesis of social inequalities. Due to the design of the study, quasi-causal analyses are also possible to a certain extent.

Apart from the fact that data users should have basic knowledge of behavioural genetic methods (for an introduction see Neale & Maes, 2004), the existing

(methodological) hurdles for working with the data are relatively low. While the scientific use file can be easily accessed via the GESIS data catalogue without any costs, extensive documentation on the study website and especially a short guide (www.twin-life.de/documentation) simplify the way for new data users to get started. The data structure, with one dataset per survey time point and a master dataset containing basic information on the participating families and their participation in the course of the study, does not pose any major challenges. With the help of existing personal and family identifiers, all data sets can be easily combined. Therefore, the data from the TwinLife study have the potential to be used for teaching to encourage interdisciplinary research among scientists.

## ADDITIONAL FILE

The additional file for this article can be found as follows:

- **Appendix.** Figures A1 to A4. DOI: https://doi.org/10.5334/jopd.78.s1

## FUNDING INFORMATION

The TwinLife study is funded by the German Research Foundation (DFG, Grant Number 220286500). The grant was awarded to Martin Diewald (DI 759/11-4), Christian Kandler (KA 4088/6-4) Rainer Riemann (RI 595/8–3), and Frank M. Spinath (SP 610/6-4).

## COMPETING INTERESTS

The authors have no competing interests to declare.

## AUTHOR AFFILIATIONS

**Theresa Rohm** orcid.org/0000-0001-9203-327X
University of Bremen, DE

**Anastasia Andreas** orcid.org/0009-0006-8600-3158
Saarland University, DE

**Marco Deppe** orcid.org/0000-0002-3927-8291
University of Bremen, DE

**Harald Eichhorn** orcid.org/0000-0003-4908-1692
Bielefeld University, DE

**Jana Instinske** orcid.org/0000-0003-0545-3554
University of Bremen, DE

**Christoph H. Klatzka** orcid.org/0000-0003-3344-8760
Saarland University, DE

**Anita Kottwitz** orcid.org/0000-0002-0341-3682
Bielefeld University, DE

**Kristina Krell**
Bielefeld University, DE

**Bastian Mönkediek** orcid.org/0000-0003-4466-715X
Bielefeld University, DE

**Lena Paulus** orcid.org/0000-0001-9660-0569
Saarland University, DE

**Sophia Piesch** orcid.org/0009-0003-1963-6586
Bielefeld University, DE

**Mirko Ruks** orcid.org/0000-0002-3621-5084
Bielefeld University, DE

**Alexandra Starr** orcid.org/0000-0002-5282-797X
Bielefeld University, DE

**Lena Weigel** orcid.org/0009-0005-2382-7865
Bielefeld University, DE

**Martin Diewald** orcid.org/0000-0002-9101-1586
Bielefeld University, DE

**Christian Kandler** orcid.org/0000-0002-9175-235X
University of Bremen, DE

**Rainer Riemann** orcid.org/0000-0001-5805-4536
Bielefeld University, DE

**Frank M. Spinath** orcid.org/0000-0002-8897-0452
Saarland University, DE

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

## PEER REVIEW COMMENTS

*Journal of Open Psychology Data* has blind peer review, which is unblinded upon article acceptance. The editorial history of this article can be downloaded here:

- **PR File 1.** Peer Review History. DOI: https://doi.org/10.5334/jopd.78.pr1

**TO CITE THIS ARTICLE:**
Rohm, T., Andreas, A., Deppe, M., Eichhorn, H., Instinske, J., Klatzka, C. H., Kottwitz, A., Krell, K., Mönkediek, B., Paulus, L., Piesch, S., Ruks, M., Starr, A., Weigel, L., Diewald, M., Kandler, C., Riemann, R., & Spinath, F. M. (2023). Data from the German TwinLife Study: Genetic and Social Origins of Educational Predictors, Processes, and Outcomes. *Journal of Open Psychology Data,* 11: 4, pp. 1–15. DOI: https://doi.org/10.5334/jopd.78

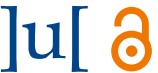