## [Peer Review History. · Journal of Open Psychology Data]

Peer Review Comments for the Data Paper “Data from the German TwinLife Study: Genetic and Social Origins of Educational Predictors, Processes, and Outcomes”

Editor:

(1) This data paper (along with several others in this collection) does not refer to an “open” dataset in the strict sense—that is data that are available to everyone and without an application process. I agree that open data would not be feasible for a study that contains sensitive information such as TwinLife. And I am also highly convinced that a research data center offering documentation, training, consultation etc. is very beneficial for researchers trying to grasp the complex data structure. And this practice of data access upon application is also typical for large panel/survey studies. Still, I think it could be made clearer that there is a transparent and clearly regulated process of data access.

In particular, it would be important to clarify if the data usage applications are evaluated based on their substantive research topics in any way and if the fact that the data depositors (as written in 3.6) have to sign off each usage application can be considered a “veto power” to reject certain applications based on the research questions. I think in the latter case it would be hard to consider the data as open. The authors would then have to make a case that either this doesn’t happen in practice (and that the agreement of data depositors is just a formality) or they need to describe the criteria for evaluation of a research proposal very clearly. It has to be confined to the readers that there is no arbitrariness in the data application process – there are rules and restrictions, but on this basis, every researcher will receive data access upon signing the data usage agreement. Along the same lines, it would be helpful to mention if there are any restrictions to data access for international researchers (within and outside the EU).

(2) I would mention a sample size in the abstract. Either per age cohort or overall, just so that the readers get a general idea of how large a study this is.

(3) In the second sentence of page 2, it is said that genetic and environmental factors affect life outcomes “beside psychological and social factor”. What does this mean exactly? What psychological factors are further predictors of life outcomes or that psychological factors are also affected by genetic and environmental factors? (or both?)

(4) On page 3, when the four age cohorts are mentioned in the manuscript for the first time (after having been mentioned in the abstract), I would name them again in parentheses because not every reader will read the abstract first.

(5) When talking about “household interviews” I would briefly mention that these can also include cognitive tests rather than just self-reports / questionnaires.

(6) I would add brief footnote related to Figure 2 explaining briefly the COVID measures in Germany. Readers will not be familiar with the term “Lockdown light” that was coined by German politicians. If a brief explanation makes it more confusing (given how many measures there were and that they varied with federal state), it could also be an option to just say “light lockdown measures” and “strict lockdown measures” or something like that.

(7) I would strongly advise to put the information presented in section 2.5 in a (large) table rather than using text and bullet points. I think that would give a better overview of the different measures. This table could either be placed in the manuscript or the appendix.

(8) The publication list presented in 2.8 could also be put in an Appendix Table.

(9) In 3.7, I would rather write “Not all data can be shared” instead of “All data cannot be shared”. Also, as mentioned above and by Reviewer B, the authors should describe the data access procedure and its limits in more detail.

(10) In section 4 it is mentioned that it would be good for users to have a basic understanding of behavioral genetic methods/statistics. Maybe it would be good to add 1-2 references to introductory works here. I imagine this very interesting dataset could inspire educational psychologists that did not work on twin data before (and thus don't have experience with the methods) to start incorporating genetic factors.

Reviewer A:

1. Being no expert on gene-environment research, there were a few expressions in the Background section (p. 2 and 3) of the manuscript which could be made clearer to readers: First, what is a genetic diathesis? Second, the difference between active and evocative gene-environment correlations could be made clearer (they are a bit mixed up, I believe). Also, dropping the “equal-environment assumption” without explaining it a bit further is a bit irritating for a reader who is not from the field.

2. In my view, the first two paragraphs could be shortened or changed to a different position within the paper, because to the reader interested in the data set, the paragraph starting with “TwinLife includes longitudinal data...” (p. 3) would be a more welcoming opening to readers who are not conducting gene-environment research.

3. On page 4, first paragraph, you state that the sample is representative with respect to educational, occupational and income characteristics, but not with regard to language background. For the reader, it would be helpful if you had a table with the sample characteristics and basic demographic information (e.g. gender, number of mono- and dizygotic twins, SES, educational background, migration background at t1) and attrition information, possibly for each cohort separately. This table would also fit in section 2.4.

4. Figure 1 could be updated to include the information that the household face-to-face interviews were substituted by CATI.

5. The subsample description is not very clear. First of all, it would be better positioned earlier in the text, because the reader is already confronted with the subsamples in Figure 2. Also, I am not sure I correctly understand the criteria for being in subsamples A and B. You write “In conclusion, each age cohort consists of two birth sub-cohorts which are surveyed in consecutive years, so that they are surveyed at the same age.” (p. 9-10). The first point of data collection was between October 2014 – April 2016, i.e. spread across 19 months and three calendar years. This is a bit confusing. If I understand correctly, subsample A1 is born in 2009 and subsample B1 in 2010. So being in subsamples A or B does not depend on the time of the first survey but only on the birth year. Maybe you could formulate that more clearly.

6. There was one detail about the sampling procedure I didn't quite understand. You write "Three subsamples of communities were then selected: a "basic sample", an "urban sample", and a "rural sample"." (p.9). Does this mean, the sample was stratified by community type and within these strata, random samples were drawn? If so, which size were these? Also, this is the place where I would expect information on weighting (which is in the section "Quality Control" (p. 20) right now).

7. You present a list of constructs very relevant to research on educational attainment. Because of growing interest in social relationships and network data, I would love to see some information on the social network data that were collected.

8. P. 20: You write that a professional survey institute conducted the surveys. Could you please specify which one?

9. Data anonymisation and ethical issues: The first paragraph is duplicated at the end of the section. Also, I am missing information on informed consent here. Who was asked for consent at which time points?

10. Formats (p. 24): You didn't give the versions of the formats used (in SPSS, for example, you find the version number when you open a .sav-file in a text viewer). Also, I believe, documentation is available in .pdf and .xlsx, you could list that here as well. You could make your data more FAIR by also using open formats, like .csv, which can be read with any text viewer (ideally together with a syntax file for importing the data to a free statistics software package like R). .dta and .sav rely on commercial software that might not be available to all users.

11. License (p. 24): You write that data is accessible with written permission from the data depositor. Who exactly is this (is that a consortium or specific persons? What happens after these persons are not available anymore? Is there a succession plan? How is it ensured that secondary researchers can continually have access to the data?) Which are the criteria for obtaining permission? Are there any transparent conditions?

12. Limits to sharing (p.25): molecular genetic samples cannot be shared due to being especially sensible data. However, if I understand the study description correctly, the genetic data would have a high scientific value and potential for re-use. Are there any possibilities to re-use these (e.g. via restricted access options) or do the informed consent sheets prohibit sharing and re-use of these data?

13. FAIR data (p.25): please also include the doi/persistent identifier of the data (if possible).

14. Possibly some parts in the manuscript that relate to the most recent waves of data collection (autumn 2022), can be updated before publication.

These seem to be quite a number of points for clarification, but I believe that many of these require only a few changes in formulations etc. The most important points that might have to be addressed to make the data really FAIR are 10 and 11. I am looking forward to the published manuscript and I believe the data have a very high potential for re-use.

Reviewer B:

page 4: The authors mention the culture fair test as an objective measure for educational achievement. However, later they describe the test as a measure for cognitive abilities, which most educational researchers would probably label as two very different things.

page 5: The term "mating" might confuse readers. Does this refer to the relationship status of study participants or having children?

page 6 & 7: Abbreviations differ between the text and figures, which hinders readability (e.g., F2F vs. Home, CATI vs. Tele).

page 8: Saliva sampling appears in the figure but is mentioned substantially later in the text. This is a bit confusing for readers.

page 10: There are switches between present and past tense, which hinder the reading flow.

page 11: What does "filtering of study items" refer to? Response or variable deletion? Or adaptive questionnaire administration?

Section 2.5.: Presentation of example items and response options is inconsistent (e.g., number of example items given; exact wordings or paraphrasing; are full sentences being used or not) and very extensive. Even though the width of information is very helpful, it might be more appropriate for an overview of the data to only give maximum one example item per construct and maybe drop the response options completely.

page 15: What does "an additional study in Thuringia" mean?

page 17, paragraph self-efficacy: omit ";

page 20: Is "implausible information is cleared" an existing English phrase?

Section 2.7.: The first and last sentence in this section are identical.

Section 3.4: Maybe the authors could add a bit information on how/if different data versions are available, as different version numbers are mentioned in the manuscript? Are all sub-data sets available in Stata and SPSS format?

Section 3.7: This section is a bit unclear/phrased ambiguously. Are there limits to sharing regarding the available data? Are there any limits to sharing the data described in section 2.5.?